# An Effective Field Theory of Magneto-Elasticity

Shashin Pavaskar, Riccardo Penco, and Ira Z. Rothstein

*Department of Physics, Carnegie Mellon University, Pittsburgh, PA 15213, USA*

We utilize the coset construction to derive the effective field theory of magnon-phonon interactions in (anti-)ferromagnetic and ferrimagnetic insulating materials. The action is used to calculate the equations of motion which generalize the Landau-Lifshitz and stress equations to allow for magneto-acoustic couplings to all orders in the fields at lowest order in the derivative expansion. We also include the symmetry breaking effects due to Zeeman, and Dzyaloshinsky-Moriya interactions. This effective theory is a toolbox for the study of magneto-elastic phenomena from first principles. As an example we use this theory to calculate the leading order contribution to the magnon decay width due to its the decay into phonons.

## CONTENTS

## I. INTRODUCTION

In this paper we utilize effective field theory (EFT) techniques to investigate magneto-elastic phenomena in insulators in the long wavelength limit. The interaction between phonons and magnons is a well developed subject. For earlier theoretical work on phonon-magnon interactions, see for instance [1–9] and for experimental work, see [10–12]. Here, we will be utilizing the coset construction [13–16] which, to the best of our knowledge, has yet to be applied to magneto-elastic systems. A primary, but not limited, goal of this paper is to set the stage for understanding the interactions of Skyrmionic with magnons and phonons [17].

Within our EFT approach, the action is completely dictated by the spontaneous symmetry breaking pattern. In the absence of gapless modes which carry conserved quantum numbers (e.g. itinerant electrons), the relevant degrees of freedom at sufficiently low energies are the Goldstone bosons associated with the spontaneously broken global symmetries. The latter act non-linearly on the Goldstone fields, and therefore are not always manifest. The coset construction [13–16] is a powerful algorithmic tool to generate an effective action for the Goldstone modes which is invariant under all the symmetries, including the ones that are realized non-linearly. The action will be organized as a derivative expansion valid up to a cutoff energy of the order of the spontaneous symmetry breaking scale. We also use this formalism to capture systematically the consequences of a small explicit breaking of certain symmetries–e.g. due to an external magnetic field, or the presence of Dzyaloshinsky-Moriya (DM) interactions among spins.

Solids break a multitude of space-time symmetries, including translations, rotations and boosts. Moreover, homogeneous and isotropic solids possess emergent internal translational and rotational symmetries (see e.g. [18–20]), which are also spontaneously broken in the ground state, as will be discussed below. We should stress that the assumption of isotropy is convenient but by no means necessary. It is straightforward to relax this assumption and consider instead a finite subgroup of rotations (for a relativistic solid, this was done for instance in [21]). The relevant symmetries and the associated generators

are given in Table I. The resulting symmetry breaking pattern is summarized in Eq. (2).

Magneto-elastic interactions are characterized by a multitude of scales, and the derivative expansion can be implemented in different ways depending upon whether or not there are hierarchies among them. We will refer to these possible choices as different *power counting schemes*. For simplicity of presentation we will make a simple choice of scales. Exploring other hierarchies can be achieved by minor variations. Our EFT approach can in principle predict a large number of effects from first principles. Here, we will only focus on a set of illustrative observables calculated in a particular power counting scheme.

*Conventions:* we will work in units such that $\hbar = 1$. Lowercase indices $a, b, c, ...$ run over $1, 2$, lowercase indices $i, j, k, ...$ run over the number of spatial dimensions, while uppercase indices $A, B, C, ...$ run over $1, 2, 3$. We use $(-, +, +, +)$ as the metric convention. Our conventions for (anti-)symmetrization of indices are $A_{(ij)} = \frac{1}{2}(A_{ij} + A_{ji})$ and $A_{[ij]} = \frac{1}{2}(A_{ij} - A_{ji})$.

## II. RELEVANT SYMMETRIES

Given the non-relativistic nature of the system we are considering, the appropriate space-time symmetry group is the Galilean group, which is comprised of time and spatial translations, spatial rotations, Galilean boosts, and total mass (or, equivalently, particle number). As we will discuss at length below, the *spontaneous* breaking of Galilean invariance, places non-trivial constraints on the dynamics of the system, which in turn enhances predictive power.

Our system also admits a number of internal symmetries including spin rotations and, if we restrict ourselves to homogeneous and isotropic systems, an emergent internal $ISO(d)$ symmetry [20] (in $d$ spatial dimensions) whose implementation will be discussed in the next section.

All these continuous symmetries and their corresponding generators are summarized in Table I. The generators satisfy an algebra whose only non-vanishing commutators are

$$
\begin{aligned}
[L_i, K_j] &= i\epsilon_{ijk}K_k, & [L_i, P_j] &= i\epsilon_{ijk}P_k, \\
[K_i, H] &= -iP_i, & [K_i, P_j] &= -iM\delta_{ij}, \\
[Q_i, T_j] &= i\epsilon_{ijk}T_k, & [Q_i, Q_j] &= i\epsilon_{ijk}Q_k \\
[S_A, S_B] &= i\epsilon_{ABC}S_C . & [L_i, L_j] &= i\epsilon_{ijk}L_k
\end{aligned}
$$
(1)

Notice in particular that the internal symmetry generators $Q_i, S_A$ and $T_i$, commute with all the generators of the Galilei group, as befits the generators of internal symmetries.

| Symmetries | Generators |
|---|---|
| Time translations: | $H$ |
| Spatial translations: | $P_i$ |
| Spatial rotations: | $L_i$ |
| Galilean boosts: | $K_i$ |
| Total mass: | $M$ |
| Spin rotations: | $S_A$ |
| Homogeneity: | $T_i$ |
| Isotropy: | $Q_i$ |

TABLE I. *Relevant symmetries of lattice of spins in three spatial dimensions in the continuum limit. Some of these symmetries may be spontaneously and/or explicitly broken.*

| Generators | Parity | Time-reversal |
|---|---|---|
| $H$ | $+$ | $-$ |
| $P_i$ | $-$ | $+$ |
| $L_i$ | $+$ | $+$ |
| $K_i$ | $-$ | $-$ |
| $M$ | $+$ | $-$ |
| $S_A$ | $+$ | $+$ |
| $T_i$ | $-$ | $+$ |
| $Q_i$ | $+$ | $+$ |

TABLE II. *Transformation properties of various symmetry generators under parity and time-reversal. Each generator $X$ in the first column transforms as $iX \to \pm iX$ with the appropriate sign shown in the second and third column.*

Discrete symmetries such as parity and time-reversal will also play an important role in what follows. The transformation properties of the above generators under these symmetries are listed in Table II. Under parity and time-reversal, each generator $X$ in the first column transforms as $iX \to \pm iX$ with the appropriate sign shown in the second and third column. A factor of "$i$" was included in these transformation rules for later convenience, to more easily account for the fact that time-reversal is implemented in a way that is anti-linear and anti-unitary (as opposed to parity, which is linear and unitary). Notice however that our transformation rules are equivalent to the ones that some readers may already be familiar with. For instance, the transformation rule of the spin $S_A$ under time reversal, which we write as $iS_A \to iS_A$, is equivalent to $S_A \to -S_A$ owing to the anti-linear nature of time-reversal.

## III. EFFECTIVE ACTIONS

In this section, we will discuss the way in which the symmetries are realized in (anti-)ferromagnets and ferrimagnets. We first address how some of these symmetries are spontaneously broken, and derive the effective action

for the ensuing Goldstone modes. A discussion of explicit symmetry breaking is postponed until Section VII.

## A. Spontaneous symmetry breaking pattern

The full symmetry group will be denoted by $G$ with elements $g$, while the unbroken subgroup will be denoted by $H$ with elements $h$. The vacuum manifold corresponds to the coset $G/H$. (Anti-)ferromagnets and ferrimagnets have the same symmetry breaking pattern save for time reversal, as depicted in Figure 1. Including lattice effects, all three cases possess the following spontaneous breaking pattern:

$$
unbroken = \begin{cases} H \\ P_i + T_i \equiv \bar{P}_i \\ L_i + Q_i \equiv \bar{L}_i \ , \\ S_3 \\ M \end{cases} \quad broken = \begin{cases} K_i \\ T_i \\ Q_i \\ S_1, S_2 \equiv S_a \end{cases} ,
$$
(2)

where we have assumed the spins to be oriented along the "3" direction. This pattern describes all the spin configurations in Figure 1. The distinction between these cases can be understood by recalling that $S_A \to -S_A$ under time-reversal. Thus, the first configuration (ferromagnets) maximally breaks time-reversal invariance, the second one (antiferromagnets) preserves it, and the last one (ferrimagnets) once again breaks it, but in a more "gentle way", as the amount of breaking is controlled by the difference between the magnitude of the spins pointing upward and those pointing downwards. In other words, time-reversal gets restored in the limit where these spins have the same magnitude. As is well known, the fate of time-reversal invariance turns out to have a significant effect on the spectrum of gapless modes (see *e.g.* [22]), which will be discussed in Section V B.

At this stage, it is worth pointing out that, since $T_i$ and $Q_i$ are spontaneously broken, $P_i$ and $L_i$ must be as well in order for the linear combinations $\bar{P}_i$ and $\bar{L}_i$ to remain unbroken. In fact, broken generators are always defined only up to the addition of unbroken ones. The broken generators listed above are just one particular choice of bases for the coset space of broken symmetries. Moreover, since some of these are space-time symmetries, not all the broken generators in our basis will give rise to Goldstone modes [23]. As we will see, phonons and magnons are the only Goldstone modes associated with the symmetry breaking pattern in Eq. (2).

## B. Coset construction for phonons and magnons

Starting from the symmetry breaking pattern (2), there exists a systematic procedure, known as the *coset construction*, [13–16] to write down a low energy effective action for the Goldstone modes. A modern and concise

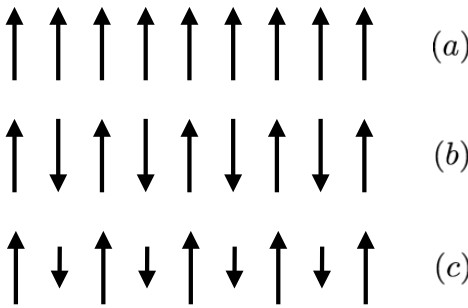

FIG. 1. Schematic representation of the ground state spin configuration of ($a$) ferromagnets, ($b$) antiferromagnets, and ($c$) ferrimagnets.

review of this technique can be found for instance in Sec. 2 of [24]. We will now apply it to the problem at hand to write down an effective action for phonons and magnons.[1]

The starting point of a coset construction is a choice of parametrization of the vacuum manifold. The parametrization that we will work with is

$$
\Omega = e^{-iHt} e^{ix^i \bar{P}_i} e^{i\eta^i K_i} e^{i\pi^i T_i} e^{i\theta^i Q_i} e^{i\chi^a S_a}.
$$
(3)

There is a considerable amount of freedom involved in choosing this parameterization, as the order of the exponentials and the basis of broken generators are to a large extent arbitrary. However, different choices are connected to each other by a field redefinition and thus generate identical predictions for physical quantities. One can think of $\Omega$ as the most general broken symmetry transformation, supplemented by an unbroken spatial and time translation.

The fields $\eta^i, \pi^i, \theta^i$ and $\chi^a$ in Eq. (3) are the Goldstone modes associated with the spontaneous breaking of $K_i, T_i, Q_i$ and $S_a$ respectively and their transformation rules under the action of $G$ is defined by the equation [16]

$$
g\,\Omega(t, x, \Phi) = \Omega(t', x', \Phi')\, h(\Phi, g),
$$
(4)

where $\Phi = \{\eta^i, \pi^i, \theta^i, \chi^a\}$, and $h$ is some element of the unbroken subgroup that generically depends on the Goldstone fields as well as the group element $g$.

As previously mentioned, not all of these modes are physically independent of each other. In fact, we will see in a moment that the fields $\eta^i$ and $\theta^i$ can be removed while preserving all the symmetries by imposing certain "inverse Higgs" constraints [30]. The remaining fields, $\pi^i$

––––––––

[1] For separate discussions of magnons and (relativistic) phonons based on the coset construction, see respectively [22, 25] and [20]. The low-energy effective theory of (anti-)ferromagnets was also discussed in [26–29].

| | $t'$ | $\vec{x}'$ | $\pi_i'(t',\vec{x}')$ | $\chi_a'(t',\vec{x}')$ |
|---|---|---|---|---|
| $H$ | $t+c$ | $\vec{x}$ | $\pi_i(t,\vec{x})$ | $\chi_a(t,\vec{x})$ |
| $\bar{P}_i$ | $t$ | $\vec{x}+\vec{a}$ | $\pi_i(t,\vec{x})$ | $\chi_a(t,\vec{x})$ |
| $\bar{L}_i$ | $t$ | $R_{ij}^{-1}(\vec{\theta})x_j$ | $R_{ij}^{-1}(\vec{\theta})\pi_j$ | $\chi_a(t,\vec{x})$ |
| $S_3$ | $t$ | $\vec{x}$ | $\pi_i(t,\vec{x})$ | $R_{ab}^{-1}(\theta_3)\chi_b(t,\vec{x})$ |
| $M$ | $t$ | $\vec{x}$ | $\pi_i(t,\vec{x})$ | $\chi_a(t,\vec{x})$ |
| $K_i$ | $t$ | $\vec{x}-\vec{v}t$ | $\pi_i+v_it$ | $\chi_a(t,\vec{x})$ |
| $Q_i$ | $t$ | $\vec{x}$ | $R_{ij}^{-1}(\vec{\theta})\phi_j-x_i$ | $\chi_a(t,\vec{x})$ |
| $T_i$ | $t$ | $\vec{x}$ | $\pi_i+c_i$ | $\chi_a(t,\vec{x})$ |
| $S_a$ | $t$ | $\vec{x}$ | $\pi_i(t,\vec{x})$ | $\chi_a(t,\vec{x})+\omega_a+...$ |
| $\mathbf{P}$ | $t$ | $-\vec{x}$ | $-\pi_i(t,\vec{x})$ | $\chi_a(t,\vec{x})$ |
| $\mathbf{T}$ | $-t$ | $\vec{x}$ | $\pi_i(t,\vec{x})$ | $\chi_a(t,\vec{x})$ |

TABLE III. *Action of the symmetries on the coordinates, the phonon fields $\pi^i$, and the magnon fields $\chi^a$.*

and $\chi^a$, will respectively describe phonon and magnon excitations. The transformation properties of coordinates, phonon fields, and magnon fields are summarized in Table III.

Starting from the coset parametrization $\Omega$, one can calculate the Maurer-Cartan form defined as $\Omega^{-1}d\Omega$:

$$\Omega^{-1}d\Omega = i\bigg\{-Hdt + \bar{P}^i(\eta^i dt + dx^i) - M(\eta^i dx^i + \tfrac{1}{2}\vec{\eta}\cdot\vec{\eta}\,dt) + Q^i\tfrac{1}{2}\epsilon^{ijk}\left[R^{-1}(\theta)dR(\theta)\right]_{jk} + K^i d\eta^i$$

$$+ T^j\left[(dx^i + d\pi^i)R^{ij}(\theta) - \eta^j dt - dx^j\right] + S^a\tfrac{1}{2}\epsilon^{aBC}\left[O^{-1}(\chi)dO(\chi)\right]^{BC}\bigg\}, \tag{5}$$

where we have introduced the matrices $R_{ij} \equiv \left(e^{i\theta^i Q_i}\right)_{ij}$ and $O_{AB} \equiv \left(e^{i\chi^a S_a}\right)_{AB}$. Note that this result follows using only the algebra in Eq. (1), and as such, can be obtained without committing to any particular representation for the group generators.

Even though we are considering a non-relativistic system, it is convenient to use the relativistic notation where $x^\mu = (t, x^i)$, and define $\bar{P}_t \equiv -H$. We should stress that this is just a matter of notational convenience, and we are not imposing Lorentz invariance. With this notation, we can rewrite the Maurer-Cartan form as follows:

$$\Omega^{-1}d\Omega \equiv idx^\nu e_\nu{}^\mu\big(\bar{P}_\mu + \nabla_\mu\pi^i T_i + \nabla_\mu\theta^i Q_i + \nabla_\mu\eta^i K_i$$
$$+ \nabla_\mu\chi^a S_a + A_\mu M + A_\mu' S_3\big). \tag{6}$$

This equation *defines* the "covariant derivatives" of the Goldstones $\nabla_\mu\pi^i, \nabla_\mu\theta^i, \nabla_\mu\eta^i$ and $\nabla_\mu\chi^a$, as well as the "connections" $A_\mu$ and $A_\mu'$ and vierbein $e_\nu{}^\mu$, which read:

$$e_0{}^0 = 1, \quad e_i{}^j = \delta_j^i, \quad e_i{}^0 = 0, \quad e_0{}^i = \eta^i \tag{7a}$$

$$\nabla_t\pi^i = (\partial_t\phi^j - \eta^k\partial_k\phi^j)R_j{}^i(\theta) \tag{7b}$$

$$\nabla_j\pi^i = \partial_j\phi^k R_k{}^i(\theta) - \delta_j^i \tag{7c}$$

$$\nabla_t\theta^i = \tfrac{1}{2}\epsilon^{ikl}\left[R^{-1}(\theta)(\partial_t - \eta^j\partial_j)R(\theta)\right]_{kl} \tag{7d}$$

$$\nabla_j\theta^i = \tfrac{1}{2}\epsilon^{ikl}\left[R^{-1}(\theta)\partial_j R(\theta)\right]_{kl} \tag{7e}$$

$$\nabla_t\eta^i = \partial_t\eta^i - \eta^j\partial_j\eta^i \tag{7f}$$

$$\nabla_j\eta^i = \partial_j\eta^i \tag{7g}$$

$$\nabla_t\chi^a = \tfrac{1}{2}\epsilon^{aBC}\left[O^{-1}(\chi)(\partial_t - \eta^j\partial_j)O(\chi)\right]_{BC} \tag{7h}$$

$$\nabla_j\chi^a = \tfrac{1}{2}\epsilon^{aBC}\left[O^{-1}(\chi)\partial_j O(\chi)\right]_{BC} \tag{7i}$$

$$A_t = \tfrac{1}{2}\vec{\eta}^2 \tag{7j}$$

$$A_i = -\eta_i \tag{7k}$$

$$A_t' = \tfrac{1}{2}\epsilon^{ab}\left[O^{-1}(\chi)(\partial_t - \eta^j\partial_j)O(\chi)\right]_{ab} \tag{7l}$$

$$A_j' = \tfrac{1}{2}\epsilon^{ab}\left[O^{-1}(\chi)\partial_j O(\chi)\right]_{ab} \tag{7m}$$

where we have defined $\phi^i \equiv x^i + \pi^i$ to streamline the notation. $\phi^i$'s are the comoving coordinates of the solid, which at equilibrium (i.e. when $\pi^i = 0$), can be chosen to be aligned with the physical coordinates $x^i$ [18].

The fields $\eta^i$ and $\theta^i$ can now be removed from the theory in a way that is compatible with all the symmetries by solving the inverse Higgs constraints [30]

$$\nabla_t\pi^i \equiv 0, \qquad\qquad \nabla_{[i}\pi_{j]} \equiv 0. \tag{8}$$

The first constraint can be solved immediately for $\eta^i$ and

yields $\eta^i = \partial_t \pi^j (D^{-1})_j{}^i$, with $D_{ij} \equiv \partial_i \phi_j$. The second constraint can instead be solved for $R_{ij}(\theta)$ using the same strategy employed for instance in Sec. V of [20]. After substituting both solutions back into the remaining covariant derivatives, the low-energy effective action will only depend on the phonon field $\pi^i$ and the magnon field $\chi^a$ through the combinations:[2]

$$\nabla_{(i}\pi_{j)} = (D\sqrt{D^T D}D^{-1})_{ij} - \delta_{ij} \tag{9a}$$

$$\nabla_t \chi^a = \frac{1}{2}\epsilon^{aBC}\left\{O^{-1}[\partial_t - \partial_t \pi^k (D^{-1})_k{}^j \partial_j]O\right\}_{BC} \tag{9b}$$

$$\nabla_i \chi^a = \frac{1}{2}\epsilon^{aBC}(O^{-1}\partial_i O)_{BC}, \tag{9c}$$

where $D_{ij} = \partial_i \phi_j = \delta_{ij} + \partial_i \pi_j$ and, once again, $O_{AB} \equiv \left(e^{i\chi^a S_a}\right)_{AB}$.

Covariant derivatives of $\eta$'s and $\theta$'s, once expressed solely in terms of the fields $\pi^i$ and $\chi^a$, turn out to have a higher number of derivatives per field compared to the ones in Eqs. (9). Thus, these quantities can be neglected at lowest order in the derivative expansion. Moreover, the coset connections $A_\mu$ and $A'_\mu$ are needed only if one is interested in higher covariant derivatives of the $\pi$'s and $\chi$'s, or in couplings with additional fields. In this paper we won't be interested in either, and therefore these connections won't play any role for our purposes.

By combining the building blocks (9) in a way that preserves the unbroken symmetries in Eq. (2), one can write down all the terms in the low-energy effective action that are *exactly* invariant under all the symmetries, including the ones that are broken spontaneously. However, the latter are realized non-linearly and thus are not

manifest. Therein lies the power of the coset construction.

There are also some terms that we can write down that are invariant only up to a total derivative. Following the high-energy physics terminology (see e.g. [31]), we will generically refer to these terms as Wess-Zumino-Witten (WZW) terms, even though they do not have a topological origin and their coefficient is not quantized. These kind of terms can be obtained systematically by combining the 1-forms that appear in front of the various generators in Eq. (5) to build 5-forms $\alpha$ that are exact, i.e. $\alpha = d\beta$, and manifestly invariant under all unbroken transformations.[3] Once again, the coset construction ensures that any $\alpha$ built this way is actually invariant under all the symmetries—including the broken ones. Therefore, the 4-form $\beta$ is in principle allowed to shift by a total derivative under a symmetry transformation [31–33], and its integral is in general a WZW term.[4] Using the solutions to the inverse Higgs constraints (8), we can always express these WZW terms solely in terms of $\pi$'s and $\chi$'s.

For the system under consideration, there are two WZW terms that we should include in our effective Lagrangian. In particular, note that if we were restricted to our building blocks (9) our action would not have time derivatives acting on the phonon field. In order to write down WZW terms, it is convenient to denote with $\omega_X$ the 1-form associated with the generator $X$ in the Maurer-Cartan form (5) up to an over all factor of "$i$". Hence, with this notation we have for instance $\omega_H = -dt$, and so on. The two exact 5-forms that we we can write down are then

$$\alpha_\pi = \epsilon_{ijk}\delta_{\ell m}\omega_{K_m} \wedge \omega_{\bar{P}_\ell} \wedge (\omega_{\bar{P}_i} + \omega_{T_i}) \wedge (\omega_{\bar{P}_j} + \omega_{T_j}) \wedge (\omega_{\bar{P}_k} + \omega_{T_k}) = d\left[\left(\eta_\ell dx^\ell + \tfrac{1}{2}\vec{\eta}^2 dt\right) \wedge d\phi^i \wedge d\phi^j \wedge d\phi^k \epsilon_{ijk}\right] \tag{10a}$$

$$\alpha_\chi = \epsilon_{ijk}\epsilon_{ab}\,\omega_{S_a} \wedge \omega_{S_b} \wedge (\omega_{\bar{P}_i} + \omega_{T_i}) \wedge (\omega_{\bar{P}_j} + \omega_{T_j}) \wedge (\omega_{\bar{P}_k} + \omega_{T_k}) = d\left[2\epsilon^{ab}(O^{-1}dO)_{ab} \wedge d\phi^i \wedge d\phi^j \wedge d\phi^k \epsilon_{ijk}\right]. \tag{10b}$$

The derivation of the RHS of Eq. (10b) is summarized in Appendix A. Once again, notice that the 5-forms above are fully invariant under all the symmetries, even though they are manifestly invariant only under the unbroken ones. The 4-forms that give rise to the relevant WZW terms are the ones in square brackets on the RHS of Eqs.

(10). Using the solutions to the inverse Higgs constraints, we can then write down the WZW terms explicitly as follows:

$$\mathcal{L}_{WZW}^\pi \equiv \frac{c_1}{2}\det(D)\left[\partial_t \pi^j (D^{-1})_j{}^i\right]^2 \tag{11a}$$

$$\mathcal{L}_{WZW}^\chi \equiv \frac{c_2}{2}\det(D)\,\epsilon^{ab}\left[(O^{-1}\partial_t O)_{ab}\right. \tag{11b}$$
$$\left. -\partial_t \pi^k (D^{-1})_k{}^j (O^{-1}\partial_j O)_{ab}\right],$$

with $c_1, c_2$ arbitrary coefficients.

Up until now we have only concerned ourselves with invariance under continuous symmetries. However, time-reversal plays a crucial role in determining the spectrum of low-energy excitations in magnetic systems. It is straightforward to derive how space-time coordinates and Goldstone fields transform under parity and

---

[2] Notice that, although it's not obvious, the tensor $(D\sqrt{D^T D}D^{-1})$ that appears in (9a) is actually symmetric. This can be checked explicitly by working perturbatively in the fields $\pi^i$.

[3] More generally, in $d$ space-time dimensions one would need to consider a $(d+1)-$form $\alpha$ that is exact.

[4] More precisely, not all the terms built this way will be WZW terms, since they could turn out to be accidentally exactly invariant. However, all WZW terms can be built this way [32].

time-reversal. To this end, we require that the coset parametrization $\Omega$ remains invariant when the broken generators transform according to the rules summarized in Table II. This leads to the transformation rules shown in Table III.

Using these results, we infer that $\nabla_{(i}\pi_{j)}, \nabla_t\chi^a$, $\mathcal{L}^\pi_{WZW}, \mathcal{L}^\chi_{WZW}$ ($\nabla_i\chi^a$) are even (odd) under parity, whereas $\nabla_{(i}\pi_{j)}, \nabla_i\chi^a, \mathcal{L}^\pi_{WZW}$, ($\nabla_t\chi^a, \mathcal{L}^\chi_{WZW}$) are even (odd) under time-reversal.

Finally, we should point out that, although the quantities in Eqs. (9) and (11) have been derived in three dimensions, they can be used in any number of spatial dimensions $d$, provided one lets the lowercase indices $i, j, k, ...$ run from 1 to $d$. In the remainder of this paper we will mostly restrict ourselves to the $d = 3$ case, unless otherwise stated.

### C. Effective action for phonons and magnons

At low-energies and large distances, the most relevant terms in the Lagrangian will be those with the least number of derivatives. In practice, this requirement means something slightly different for the phonon field $\pi^i$ and the magnon field $\chi^a$, i.e. the derivative expansion is implemented differently on the two fields. This can be easily seen from the fact that, unlike the $\chi$'s, each $\pi$ in Eqs. (9) and (11) appears with a derivative.[5] Therefore at lowest order in the derivative expansion, anharmonic corrections to the free Lagrangian for phonons and magnons are suppressed by higher powers of $\partial_i\pi^j$ and $\chi^a$ (which, with our conventions, are both dimensionless). When these quantities are small, one can safely expand the terms in Eqs.

(9) and (11) in powers of $\partial\pi$ and $\chi$ and keep only the first few terms. This is certainly the appropriate thing to do if we are interested in studying small fluctuations around a particular ground state of the system—as we will do for instance in Secs. IV B and V B.

It is however not necessary to perform such an expansion at this stage. In fact, by keeping intact the non-linear structures in (9) and (11) we will be able to also describe non-trivial field configurations where the first derivative of the phonon field is of order one, with second derivatives being suppressed. A similar approach is taken in General Relativity where the Einstein-Hilbert action can be derived starting from spin-2 perturbations around a particular ground state—the Minkowski vacuum—and then resumming all non-linear interactions that are dictated by symmetry, locality, and self-consistency [34]. This action can then be used to describe spacetimes other than Minkowski as long as higher derivative curvature invariants for these solutions remain small in units of the cutoff.

Since magnons do not carry one derivative per field, we allow the field itself to vary at the order one level, but its first derivatives must remain small in units of the cutoff. We can systematically include higher derivative corrections at the cost of introducing additional unknown Wilson coefficients.

Thus, we are going to use the full expression for our Goldstone covariant derivatives and WZW terms, and write down the most general effective Lagrangian that contains one derivative on each $\pi$, and the least possible number of derivatives on the $\chi$'s. For ferromagnets, this requirement leads to the following effective Lagrangian:

$$\mathcal{L}_{\text{ferromagnets}} = \mathcal{L}^\pi_{WZW} + \mathcal{L}^\chi_{WZW} - F_1(u) - \tfrac{1}{2}F_2^{ij}(u)\,\nabla_i\chi_a\nabla_j\chi^a, \tag{12}$$

where we have defined $u_{ij} \equiv \nabla_{(i}\pi_{j)}$ for notational convenience, $F_1$ and $F_2^{ij}$ admit an *a priori* arbitrary series expansion in powers of $u_{ij}$. Notice that the $i$-type indices and $a$-type indices cannot be contracted with each other, because the former transform under $\bar{L}_i$, whereas the latter under $S_3$. Moreover, we have not included a term of the form $\nabla_t\chi_a\nabla_t\chi^a$ which would contain a term quadratic in $\chi$ with two time derivatives, because for ferromagnets it is subleading compared to $\mathcal{L}^\chi_{WZW}$ which contains a quadratic term with only one time derivative. The latter, in turn, is allowed only because time-reversal is broken. Hence, this term cannot appear in the effective Lagrangian for anti-ferromagnets, which reads:

$$\mathcal{L}_{\text{antiferromagnets}} = \mathcal{L}^\pi_{WZW} - F_1(u) - \tfrac{1}{2}F_2^{ij}(u)\nabla_i\chi_a\nabla_j\chi^a + \tfrac{1}{2}F_3(u)\nabla_t\chi_a\nabla_t\chi^a. \tag{13}$$

The leading kinetic term for the $\chi$'s now comes from the last term in Eq. (13) rather than from $\mathcal{L}^\chi_{WZW}$, and this leads to a different dispersion relation for magnons [22], as we will see in a moment.

Finally, the low-energy excitations in ferrimagnets derive their kinetic term from an interplay between the term $\nabla_t\chi_a\nabla_t\chi^a$ and $\mathcal{L}^\chi_{WZW}$. The coefficient $c_2$ in $\mathcal{L}^\chi_{WZW}$ is much smaller than in ferromagnets since its size is determined by the scale at which time reversal is spontaneously broken, which in ferrimagnets is parametrically smaller than the scale at which all other symmetries are broken. Thus, the effective action for ferrimagnets is:

$$\mathcal{L}_{\text{ferrimagnets}} = \mathcal{L}^\pi_{WZW} + \mathcal{L}^\chi_{WZW} - F_1(u) - \tfrac{1}{2}F_2^{ij}(u)\nabla_i\chi_a\nabla_j\chi^a + \tfrac{1}{2}F_3(u)\nabla_t\chi_a\nabla_t\chi^a. \tag{14}$$

---

[5] The reason for this is that the phonons are associated with a broken Abelian group.

## IV. PHONONS

Let us start by turning off the magnon field and focusing on the phonons. Then, our effective Lagrangian reduces to

$$\mathcal{L} \to \frac{c_1}{2} \det(D) \left[\partial_t \phi^j (D^{-1})_j{}^i\right]^2 - F_1(u), \qquad (15)$$

where, as the reader may remember, we have previously defined $D_{ij} = \partial_i \phi_j$ and $u_{ij} = (D\sqrt{D^T D}D^{-1})_{ij} - \delta_{ij}$.

### A. The Elasticity equations

It is convenient to exploit the fact that, in an isotropic system, the function $F_1$ depends only on the $SO(3)$-invariant contraction of the tensor $u_{ij}$. In any such contraction, the outermost tensors $D$ and $D^{-1}$ drop out. This means that $F_1$ can also be regarded as an arbitrary function of $\sqrt{D^T D}$ or, equivalently, $(D^T D)_{ij} = \partial_k \phi_i \partial^k \phi_j \equiv B_{ij}$, which is the metric in the co-moving coordinate system. Therefore, we can work with the Lagrangian

$$\mathcal{L} \to \frac{c_1}{2} \det(D) \left[\partial_t \phi^j (D^{-1})_j{}^i\right]^2 - F_1(B), \qquad (16)$$

where, with a slight abuse of notation, we have replaced $F_1(u) \to F_1(B)$.

This action admits a simple physical interpretation if we think of the $\phi^i$'s as comoving coordinates—meaning that $\phi^i(x)$ labels the volume element at position $x$. Denoting by $\rho(\phi^i)$ the mass density in the comoving frame, the mass density in the lab frame is [18]

$$\rho(x) = \rho(\phi^i) \det(\partial_i \phi_j). \qquad (17)$$

This quantity is actually the zero component of the identically conserved current[6]

$$J^\mu = \frac{\rho(\phi)}{3!} \epsilon^{\mu\nu\rho\sigma} \partial_\nu \phi^i \partial_\rho \phi^j \partial_\sigma \phi^k \epsilon_{ijk}. \qquad (18)$$

From this current, we can deduce the velocity at which volume elements move around in the lab frame:

$$v^i = \frac{J^i}{J^0} = -(\partial_t \phi^j)(D^{-1})_j{}^i. \qquad (19)$$

With this identification, the equation $\partial_\mu J^\mu = 0$ reproduces the standard continuity equation, $\partial_t \rho + \partial_i(\rho v^i) = 0$. Notice that this result for $v^i$ is consistent with the covariant derivative in eq. (9b), where the time derivative becomes the "fisherman derivative".

Moreover, homogeneity implies that the comoving mass density must be a constant, i.e. $\rho(\phi^i) = \bar{\rho}$. This

----

[6] By identically conserved we mean that this is not a Noether current that follows from a symmetry of the Lagrangian (16).

can be deduced more formally by noting that the symmetry generators $T_i$ act on the fields $\phi^i$ as constant shifts: $\phi^i \to \phi^i + c^i$. As a result, we see that the first term in the Lagrangian (16) is just the usual kinetic energy $\frac{1}{2}\rho v^2$ with the identification $c_1 \equiv \bar{\rho}$; the second term can be thought of as a potential energy contribution.

The equations of motion can be obtained as usual from the Euler-Lagrange equations for $\pi^i$, or equivalently $\phi^i$, that follow from the Lagrangian (16). However, as is usually the case for Goldstone fields, their equation of motion are also equivalent to the conservation equations for the associated broken generators. In our case, the equations for the phonons follow from the conservation equations for the "homogeneity generators" $T_i$. Equivalently, we can also consider the equations for momentum conservation, since the momentum generators $P_i$ and the $T_i$'s are equivalent up to an unbroken generator: $T_i = \bar{P}_i - P_i$. We therefore consider

$$\partial_\mu T^{\mu i} = 0, \qquad (20)$$

with

$$T^{\mu i} = \frac{\partial L}{\partial(\partial_\mu \phi^j)} \partial^i \phi^j - \eta^{\mu i} L. \qquad (21)$$

An explicit calculation of $T^{\mu i}$ yields

$$T^{0i} = \bar{\rho}(\det D)(\partial_t \phi^k (D^{-1})_k^i) = -\rho v^i \qquad (22a)$$

$$T^{ij} = \frac{\partial L}{\partial D^{ik}} D^{jk} - \delta^{ij} L = -\rho v^i v^j + \sigma^{ij}. \qquad (22b)$$

where we have identified the *stress tensor*

$$\sigma_{ij} \equiv \tilde{F}_1 \delta_{ij} - 2 \frac{\partial \tilde{F}_1}{\partial B^{k\ell}} \partial_i \phi^k \partial_j \phi^\ell. \qquad (23)$$

Then, leveraging the conservation of the current (18), Eq. (20) reduces to the familiar elasticity equations:

$$\rho(\partial_t + v^j \partial_j) v^i = \partial_j \sigma^{ji}, \qquad (24)$$

### B. Phonon Spectrum

Let us now expand the Lagrangian (16) up to quadratic order in the $\pi$ fields to derive the existence of phonon excitations in the static unstressed ground state $\langle \phi^I \rangle = x^I$. Expanding $B_{ij}$ in the phonon fields $\pi$'s, we find

$$B_{ij} = \delta_{ij} + \partial_i \pi_j + \partial_j \pi_i + \partial_k \pi_i \partial^k \pi_j \qquad (25)$$

At quadratic order in the $\pi$ fields the Lagrangian is then given by

$$\mathcal{L}_\pi^{(2)} = \frac{c_1}{2} \partial_t \pi^i \partial_t \pi_i - \frac{c_4 + c_5}{2} (\partial_i \pi^i)^2 - \frac{c_5 + c_3}{2} \partial_i \pi_j \partial^i \pi^j \quad (26)$$

where the coefficients $c_3, c_4$ and $c_5$ are defined by the relations:

$$\frac{\partial F_1}{\partial B^{ij}} \bigg|_{\delta_{ij}} \equiv \frac{c_3}{2} \delta_{ij} \qquad (27)$$

$$\frac{\partial^2 F_1}{\partial B^{ij} \partial B^{kl}} \bigg|_{\delta_{ij}} \equiv \frac{c_4}{4} \delta_{ij} \delta_{kl} + \frac{c_5}{4} (\delta_{ik} \delta_{jl} + \delta_{jk} \delta_{il}). \qquad (28)$$

where we have utilized the isotropy of the background. Given the assumption of isotropy, we can decompose the strains into their irreducible components

$$\partial_i \pi_j = (S_{ijkl} + A_{ijkl} + T_{ijkl})\partial_k \pi_l \qquad (29)$$

where $S_{ijkl}, A_{ijkl}$ and $T_{ijkl}$ are the projectors onto the symmetric-traceless, anti-symmetric and the trace parts.

$$S_{ijkl} = \frac{1}{2}(\delta_{ik}\delta_{jl} + \delta_{il}\delta_{jk}) - \frac{1}{3}\delta_{ij}\delta_{kl}$$
$$A_{ijkl} = \frac{1}{2}(\delta_{ik}\delta_{jl} - \delta_{il}\delta_{jk}) \qquad (30)$$
$$T_{ijkl} = \frac{1}{3}\delta_{ij}\delta_{kl}$$

It is easy to see that the anti-symmetric part is just the $\theta$ goldstone and can be set to zero since we have integrated it out. The irreducible components of the strains are orthogonal to each other. The decomposition in (29) allows us to re-write the action in (26) as

$$\mathcal{L} = \frac{c_1}{2}(\partial_t \pi^i)^2 - \frac{c_5 + c_3}{2}(S_{ijkl}\partial^k \pi^l)^2 - \frac{4c_5 + 3c_4 + c_3}{2}(T_{ijkl}\partial^k \pi^l)^2 \qquad (31)$$

This puts constraints on the coefficients of the Lagrangian

$$G \equiv c_5 + c_3 > 0 \quad 3K \equiv 2c_5 + 3c_4 - c_3 > 0 \qquad (32)$$

where we have identified the coefficients with the shear $G$ and bulk modulus $K$. This is straightforward to see since the trace part only contributes to pure compression whereas the traceless symmetric part contributes to pure shear of the material. It is now convenient to decompose $\pi^i$ into the sum of a longitudinal part $\pi_L^i$ and a transverse part $\pi_T^i$, such that

$$\vec{\nabla} \cdot \vec{\pi}_T = 0, \qquad \vec{\nabla} \times \vec{\pi}_L = 0. \qquad (33)$$

It follows from the Lagrangian (31) that these two components satisfy two different wave equations, which admit solutions—the sound waves, or phonons—with linear dispersion relations $\omega^2 = v_{L,T}^2 k^2$, and longitudinal and transverse speeds given by

$$v_L^2 = \frac{4G + 3K}{3\bar{\rho}} \qquad v_T^2 = \frac{G}{\bar{\rho}} \qquad (34)$$

From (32), this implies that $v_L^2 > \frac{4}{3}v_T^2$.[7]

---

[7] See however [35] for an interesting UV model that violates this bound.

## C. Power Counting

The effective Lagrangian (16) is the leading term in a suitably defined derivative expansion. This means that the elasticity equations we derived from it are only valid to the extent that higher derivative corrections are negligible. Similarly, the quadratic Lagrangian (26) can be trusted only if it is safe to neglect the non-linear corrections that arise by expanding (16) to higher orders in $\pi^i$. Under what circumstances are these good approximations?

To address this question, we will make the simplifying assumption that $v_L$ and $v_T$ are of the same order, which we will schematically denote with $v_\pi$. Then, the effective action (16) can be written as

$$\frac{S}{\hbar} = \int dt d^3r \, \frac{\bar{\rho} v_\pi^2}{\hbar} \mathcal{L}(\dot{\pi}/v_\pi, \partial_i \pi_j), \qquad (35)$$

where we have momentarily reintroduced an explicit factor of $\hbar$ to make dimensional analysis more transparent. On naturalness grounds, we will assume that the Lagrangian density $\mathcal{L}$—which is a dimensionless function of dimensionless arguments—only contains coefficients of order one. This implies immediately that quadratic Lagrangian (26) is a good approximation for field configurations such that $\dot{\pi}/v_\pi, \partial_i \pi_j \ll 1$.

It is convenient to introduce a new time variable $t' \equiv v_\pi t$. This is equivalent to introducing new units such that time is measured in the same units as lengths, and the sound speeds are dimensionless numbers of $\mathcal{O}(1)$. In these new units, the action above becomes

$$\frac{S}{\hbar} = \int dt' d^3r \, \frac{\bar{\rho} v_\pi}{\hbar} \mathcal{L}(\partial_{t'}\pi, \partial_i \pi_j). \qquad (36)$$

This action now depends on a single length scale, $L_\pi \equiv (\bar{\rho} v_\pi/\hbar)^{-1/4}$, which therefore should be identified with the length cutoff of our effective theory. This means that higher derivative corrections to (36) must appear in the combinations $L_\pi \partial_i$ and $L_\pi \partial_{t'} = (L_\pi/v_\pi)\partial_t$. Hence, our effective action can reliably describe phonon excitations with frequencies $\omega \ll v_\pi/L_\pi$ and wave-numbers $|\vec{k}| \ll 1/L_\pi$.

## V. MAGNONS

In the incompressible limit one can neglect the phonon field, and the effective Lagrangian for the magnon fields reduces to

$$\mathcal{L} \to \frac{c_2}{2}\epsilon^{ab}(O^{-1}\partial_t O)_{ab} + \frac{c_6}{2}(\nabla_t \chi_a)^2 - \frac{c_7}{2}(\nabla_i \chi_a)^2, \quad (37)$$

where we have defined $F_3(u = 0) \equiv c_6$ and $F_2^{ij}(u = 0) \equiv c_7 \delta^{ij}$. The coefficient $c_2$ is $\sim (c_6 c_7)^{3/4}$ for ferromagnets, $\ll (c_6 c_7)^{3/4}$ for ferrimagnets, and vanishes for antiferromagnets.

## A. Nonlinear Equations of Motion

As we did for the phonons in the previous section, we can easily derive the non-linear equations of motion for the magnons. This will allow us to make contact with the standard literature on magnetism. To this end, it is convenient to perform the following field redefinition:

$$\chi_1 \equiv \theta \sin \phi, \qquad \chi_2 \equiv -\theta \cos \phi, \qquad (38)$$

and to introduce the unit-norm vector

$$\hat{n} = O(\chi)\hat{x}_3 = (\sin\theta\cos\phi, \sin\theta\sin\phi, \cos\theta). \qquad (39)$$

In terms of these new fields, after some algebra, the Lagrangian (37) becomes

$$\mathcal{L} \to - c_2\, \dot{\phi} \cos\theta + \tfrac{c_6}{2}(\partial_t \hat{n})^2 - \tfrac{c_7}{2}(\partial_i \hat{n})^2. \qquad (40)$$

Note that the first term doesn't admit a simple expression in terms of $\hat{n}$ because, unlike the other ones, it is only invariant up to a total derivative. This can be easily checked using the fact that $\hat{n}$ transform linearly under spin rotations, and hence that its change under infinitesimal spin rotations is $\delta\hat{n} = \vec{\omega} \times \hat{n}$. This implies that

$$\delta\theta = \omega_y \cos\phi - \omega_x \sin\phi$$
$$\delta\phi = \omega_z - \omega_x \cot\theta\cos\phi - \omega_y \cot\theta\sin\phi, \qquad (41)$$

or, equivalently, that the $\chi$ fields must transform as

$$\delta\chi_1 = -\frac{\omega_x}{1+\chi_1^2/\chi_2^2}(\chi_1^2/\chi_2^2 + \sqrt{\chi_1^2+\chi_2^2}\cot\sqrt{\chi_1^2+\chi_2^2})$$
$$\delta\chi_2 = -\frac{\omega_y}{1+\chi_1^2/\chi_2^2}(1+\chi_1^2/\chi_2^2\sqrt{\chi_1^2+\chi_2^2}\cot\sqrt{\chi_1^2+\chi_2^2}). \qquad (42)$$

It is then easy to check that the Lagrangian (40) changes by a total time derivative under a spin rotation:

$$\delta\mathcal{L} = -\frac{d}{dt}\left[\frac{1}{\sin\theta}\left(\omega_y\sin\phi + \omega_x\cos\phi\right)\right]. \qquad (43)$$

Once again, rather than deriving the equations of motion by varying the Lagrangian (37) with respect to our fields, we will resort to the conservation of the Noether currents associated with spin rotations. In order to calculate the currents, we must account for the fact that the WZ term is only invariant up to a total time derivative. Including this contribution leads to

$$J_a^\mu = (-n_a, (\vec{\nabla}n \times \hat{n})_a). \qquad (44)$$

The equations of motion, for $\theta$ and $\phi$ can now be written in a very compact form in terms of $\hat{n}$ by imposing $\partial_\mu J_a^\mu = 0$ to find:

$$c_2\, \partial_t \hat{n} = -(c_6\partial_t^2\hat{n} - c_7\nabla^2\hat{n}) \times \hat{n}. \qquad (45)$$

When $c_6\partial_t \ll c_2$, the first term on the righthand side can be neglected, and our result reduces to the well-known *Landau-Lifshitz equation* for ferromagnets [22, 36].

The informed reader will notice that these equations are missing the so-called "Gilbert damping" term, induced by the magnon finite lifetime. As is well known, an action formalism, from which we have derived our equations of motion, is inherently time symmetric. To account for damping one should work within the so-called "in-in" formalism. In section (VI D) we will calculate the magnon damping using our formalism. To generate the Gilbert damping would entail using these results in conjunction with the in-in formalism [37].

## B. Magnon Spectrum

Let us now turn our attention to the spectrum of long-wavelength excitations around the ground state. For simplicity, we will work with the Lagrangian (37), which strictly speaking is appropriate for ferrimagnets; (anti-)ferromagnets can be easily recovered by taking appropriate limits. These limits will in turn affect the power counting, as we will discuss in the next section.

Expanding (37) up to quadratic order in the $\chi$'s, we find

$$\mathcal{L}_\chi^{(2)} = \frac{c_2}{2}\epsilon_{ab}\chi^a\partial_t\chi^b + \frac{c_6}{2}\partial_t\chi_a\partial_t\chi^a - \frac{c_7}{2}\partial_i\chi_a\partial^i\chi^a, \quad (46)$$

The dispersion relations for the magnon modes then follow by demanding that the determinant of the quadratic kernel vanishes in Fourier space. If the coefficient $c_2$ doesn't vanish, as is the case for ferri- and ferro-magnets, then one finds that, in the small $k$ limit,

$$\omega_+^2 \simeq \Delta^2 + \mathcal{O}(k^2), \qquad \omega_-^2 \simeq \left(\frac{k^2}{2m}\right)^2 + \mathcal{O}(k^6), \quad (47)$$

where we have introduced the gap $\Delta = c_2/c_6$ and the effective mass $m = c_2/(2c_7)$. The gapped modes with dispersion relation $\omega_+^2$ are physical provided $c_2$ is small enough that the energy gap $\Delta$ falls below the cutoff of the effective theory. This is the case for ferrimagnets, but not ferromagnets, as we discuss in the following section and further elaborate on in Appendix B.

When $c_2 = 0$, one instead finds two modes with identical linear dispersion relation:

$$\omega_\pm^2 = v_\chi^2 k^2, \qquad (48)$$

with the phase velocity equal to $v_\chi^2 = c_7/c_6$. Note that the three parameters that appear in the dispersion relations above are not all independent: they are related to each other by $\Delta = 2mv_\chi^2$. The mechanism by which a term with a single time derivatives can turn a pair of gapless modes with linear dispersion relation into a gapped mode and a mode with quadratic dispersion relation has been studied extensively in the literature—see e.g. [38–40] and references therein.

### C. Power counting

Let us first consider anti-ferromagnets, where $c_2 = 0$; in this case, the low-energy effective Lagrangian (37) acquires an accidental symmetry. Although Galilean boosts appear to be explicitly broken in the incompressible limit, when the phonon fields are neglected, the Lagrangian for antiferromagnets is formally invariant under Lorentz transformations with "speed of light" $v_\chi^2 = c_7/c_6$; indeed, it has the same form as the Lagrangian for a relativistic nonlinear sigma model $SO(3)/SO(2)$. This additional symmetry ensures that the coefficients $c_{6,7}$ get renormalized by nonlinearities in (37) in such a way that their ratio remains constant. Higher derivative corrections to (37) won't generically preserve this accidental symmetry—even though it would be technically natural for them to do so—and can therefore affect the ratio $c_7/c_6$.

Because of this accidental symmetry, the power counting scheme for anti-ferromagnets is virtually identical to that for a relativistic theory, with the speed of light replaced by $v_\chi$. Keeping length and time scales separate, we find that the only length scale that can be built out of $c_6$ and $c_7$ is $L_\chi = (c_6 c_7)^{-1/4}$, and the only time scale is $L_\chi/v_\chi$. In the absence of fine-tunings, these must be the scales that suppress higher derivative corrections to the effective Lagrangian (37) (as usual, up to loop factors of $4\pi$ and coefficients of order one).[8] In other words, observables in the effective theory can be calculated in an expansion in powers of $\omega L_\chi/v_\chi$ and $k L_\chi$. Furthermore, non-linearities in (37) are suppressed compared to the quadratic terms as long as $\chi^a \ll 1$.

Let us now turn our attention to the case of ferromagnets, where $c_2 \sim L_\chi^{-3}$. The gap $\Delta$ becomes comparable to the energy cutoff of the effective theory, i.e. $\Delta \sim v_\chi/L_\chi$ [9], and therefore the corresponding mode exits the regime of validity of the effective theory. An equivalent viewpoint is that the second term in the quadratic Lagrangian (46) becomes negligible compared to the first one for $\omega \ll v_\chi/L_\chi$. By themselves, the first and third term describe a single propagating mode with a non-relativistic dispersion relation—the second mode in Eq. (47). In fact, combining the $\chi^a$ in a single complex field $\Psi = \chi_1 + i\chi_2$, the Lagrangian (46) with $c_6 = 0$ reduces to the standard Lagrangian for a non-relativistic field $\Psi$. Thus, in this case the power counting is implemented exactly like in a theory for non-relativistic point particles (see e.g. [24, 41]).[10]

Finally, let us discuss the case of ferrimagnets, where $c_2$ is non-zero but small in units of the cutoff, i.e. $c_2 L_\chi^3 \ll 1$. This ratio introduces an additional expansion parameter that controls the soft breaking of time reversal [25]. The low-energy excitations are akin to a light relativistic

particle and a heavy non-relativistic particle interacting with each other (of course, the interactions that are not invariant under Galilei nor Lorentz boosts). At energies $\Delta \ll \omega \ll v_\chi/L_\chi$, the gap is negligible and one is left with an essentially gapless mode interacting with a heavy non-relativistic particle; explicit power counting can then be implemented as in non-relativistic QED and QCD [24, 41, 42]. At energies $\omega \ll \Delta$, one can treat also the gapped mode as non-relativistic, and switch to a new effective theory with cutoff $\Delta$ that describes soft interactions of two non-relativistic particles with widely separated masses $\Delta$ and $m$. Note that there is no distinction between the various cases

As in the case of the solid we may relate the cut-off to the UV parameters of the theory. There is one fundamental energy scale $J$, the exchange energy (see section VII ) and one length scale, the lattice spacing $a$. Therefore, these must be the length ($L_\chi = a$) and time ($L_\chi/v_\chi = \hbar/J$) scales which suppress higher dimensional operators.

## VI. MAGNON-PHONON INTERACTIONS

We will finally turn our attention to the coupled system of phonons and magnons. Magnetoelastic effects have already been studied in ferromagnets [9, 43–46], ferrimagnets [47], and antiferromagnets [48–50]. However, the focus has been on particular effects (e.g. Spin Seebeck effect [51–53]) or particular materials (e.g. Yttrium Iron Garnet [11, 46]). In contrast, we are interested in universal low-energy phenomena that follow directly from symmetries. In this section we will derive a few such results.

### A. Generalized equations of motion

We will start by deriving the coupled equations of motion for magnon and phonon fields, which generalize the elasticity and Landau-Lifshitz equations discuss previously. In order to obtain the most general form of these equations, we work with the Lagrangian for ferrimagnets. Using the definition for $\hat{n}$, $\rho$ and $\vec{v}$ we can rewrite Eq. (14) as

$$\mathcal{L} = \tfrac{1}{2}\rho v^2 + \mathcal{L}_{WZW}^\chi - F_1(B) \qquad (49)$$
$$- \tfrac{1}{2} F_2^{ij}(B)\, \partial_i \hat{n} \cdot \partial_j \hat{n} + \tfrac{1}{2}\rho \tilde{F}_3(B) D_t \hat{n} \cdot D_t \hat{n}$$

where in the last term we have used eq. (9b) and defined $D_t \equiv (\partial_t + v^i \partial_i)$ and redefined $F_3 = \rho \tilde{F}_3$. Varying this Lagrangian with respect to the magnon fields, we obtain

---

[8] Of course, one can always *engineer* materials where this assumption fails, i.e. higher derivative terms are suppressed by unnaturally small coefficients. In this case, the power counting must be

adjusted accordingly.

$$\rho \frac{c_2}{c_1} D_t \hat{n} - \rho \hat{n} \times D_t(\tilde{F}_3 D_t \hat{n}) + \rho \tilde{F}_3 \partial_i v^i \hat{n} \times D_t \hat{n} + \hat{n} \times \partial_i(F_2^{ij} \partial_j \hat{n}) = 0, \tag{50}$$

while varying with respect to the phonon fields yields:

$$\rho\, D_t \left[ v_i + \frac{c_2}{2c_1} \epsilon^{ab}(O^{-1}\partial_i O)_{ab} + \tilde{F}_3 D_t \hat{n} \cdot \partial_i \hat{n} \right] = \partial_j(\sigma_{ji} + \bar{\sigma}_{ji} + \tilde{\sigma}_{ji}), \tag{51}$$

where

$$\bar{\sigma}_{ji} = (\partial_m \hat{n}) \cdot (\partial_p \hat{n}) \left[ \frac{\delta_{ij}}{2} F_2^{mp} - \frac{\partial F_2^{mp}}{\partial B^{kl}} \partial_k \phi_i \partial_l \phi_j \right] \tag{52}$$

$$\tilde{\sigma}_{ji} = -\rho(D_t\hat{n}) \cdot (D_t\hat{n}) \left[ \frac{\delta_{ij}}{2} \tilde{F}_3 - \frac{\partial \tilde{F}_3}{\partial B_{lk}} \partial_i \phi_l \partial_j \phi_k \right]. \tag{53}$$

These equations are a generalization of previous works on magneto-elastic equations [54–57]. Notice that we have used the continuity equation $\partial_t \rho + \partial_i(\rho v^i) = 0$ to simplify Eqs. (50) and (51). We can recover the equations for ferromagnets (anti-ferromagnets) by setting $\tilde{F}_3 = 0$ ($c_2 = 0$). Interestingly, when the stresses on the right-hand side of Eq. (51) are negligible, the quantity that is conserved in a comoving sense is no longer the local velocity of the solid, but in fact a combination that also involves the magnons. To the best of our knowledge the results for the fully non-linear equations of motion, to leading order in derivatives, (50) and (51) are novel.

### B. Power Counting in the Mixed Theory

Once we consider both magnons and phonons at the same time, the power counting becomes much more complex. Consider, for instance, the case of antiferromagnets, for which $\mathcal{L}_{WZW}^\chi = 0$. We now have two characteristic length scales, $L_\chi$ and $L_\pi$ (which need not be of the same order as their ratio is dictated by the microphysics), and at least two independent speeds, $v_\chi$ and $v_\pi$ (assuming that longitudinal and transverse speeds are of the same order, which need not be the case). Based on our previous discussions on power counting, the natural expectation is that the functions appearing in the Lagrangian (49) scale like

$$F_1 \sim \frac{v_\pi}{L_\pi^4}, \qquad F_2^{ij} \sim \frac{v_\chi}{L_\chi^2}, \qquad \tilde{F}_3 \sim \frac{v_\pi L_\pi^4}{v_\chi L_\chi^2}, \tag{54}$$

and that higher powers of $\dot{\pi}$ are suppressed by $v_\pi$. Observables should now be calculated in an expansion in powers of $\omega L_</v_>$, $kL_<$, $L_</L_>$, and $v_</v_>$, where $L_>$

($L_<$) is the largest (smallest) between $L_\pi$ and $L_\chi$, and similarly for the speeds.

Unfortunately, one cannot associate *a priori* a definite scaling to each term in the Lagrangian (49). This is because, when vertices are combined into Feynman diagrams, internal lines can be off-shell but an amount that is controlled by one or more of the expansion parameters listed above. A similar problem occurs in non-relativistic QED and QCD, and it's handled by resorting to the method of regions (see e.g. [24, 41, 42, 58]). Ferro- and ferri-magnets[11] presents a similar challenge, except that the relevant kinematical regions are different compared to those of ferromagnets.

Ultimately, these subtleties related to power counting become relevant only if one wants to calculate higher order corrections in a systematic way. At lowest order, it is usually straightforward to drop subleading corrections and zero in on the leading contribution to whatever process one is interested in. To illustrate this, in what follows we will consider the leading corrections to the propagation of magnons due to couplings with the phonons. At leading order, these effects are captured by interactions in the Lagrangian (49) that are quadratic in $\chi$ and linear in $\pi$

$$\begin{aligned} \mathcal{L}_{\rm int} = & \frac{c_2}{2} \partial_i \pi^i \epsilon_{ab} \chi^a \partial_t \chi^b - \frac{c_2}{2} \epsilon_{ab} \chi^a \partial_t \pi^i \partial_i \chi^b \\ & - \frac{c_8}{2} \partial_k \pi^k \partial_i \chi^a \partial^i \chi^a - c_9 \partial_i \chi^a \partial_j \chi^a \partial^{(i} \pi^{j)} \\ & + \frac{c_{10}}{2}(\partial_i \pi^i)(\partial_t \chi^a)^2 - c_6 \dot{\chi}_a \dot{\pi}^k \partial_k \chi_a, \end{aligned} \tag{55}$$

where we have defined

$$\frac{\delta F_2^{ij}}{\delta B_{kl}} \equiv \frac{c_8}{2}\delta_{ij}\delta_{kl} + \frac{c_9}{2}(\delta_{ik}\delta_{jl} + \delta_{il}\delta_{jk}), \tag{56a}$$

$$\frac{\delta F_3}{\delta B_{ij}} \equiv \frac{c_{10}}{2}\delta_{ij}. \tag{56b}$$

It is straightforward to estimate the natural size of the coefficients in (55) in terms of $L_{\chi,\pi}$ and $v_{\chi,\pi}$.

### C. Magnons in a stressed sample

Consider now a magnetic material under the application of a constant stress (normal and shear). This causes

---

[9] Here we have used the relation $c_2 \sim (c_6 c_7)^{3/4}$ valid for ferromagnets.

[10] One technical difference compared to ordinary non-relativistic particles is that all magnon self-interactions are suppressed by at least two derivatives.

[11] As we discussed in the previous section, ferrimagnets feature yet another expansion parameter, $c_2 L_\chi^3$, controlling the amount of time reversal breaking.

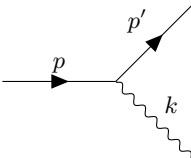

FIG. 2. Feynman diagram describing the emission of a phonon from a magnon

the atoms to displace from their equilibrium positions, which is captured by a non-zero expectation value for the phonon fields. We will denote the linear strain tensor in the sample by $\gamma_{ij} = \langle \partial_{(i} \pi_{j)} \rangle$. In the limit where the strain is small (note that $\gamma_{ij}$ is dimensionless), the leading corrections to the quadratic Lagrangian for magnons in Eq. (46) will come from the interactions shown in Eq. (55) with the phonon fields replaced by their expectation value:

$$\mathcal{L}_{\text{int}} \to \frac{c_2 \gamma}{2} \epsilon_{ab} \chi^a \partial_t \chi^b - \frac{c_8 \gamma}{2} \partial_i \chi^a \partial^i \chi^a \tag{57}$$
$$- c_9 \gamma^{ij} \partial_i \chi^a \partial_j \chi^a + \frac{c_{10} \gamma}{2} (\partial_t \chi^a)^2,$$

where we used the fact that the shear is by assumption time-independent, and we defined $\gamma = \delta^{ij} \gamma_{ij}$.

Assuming moreover that the stress is homogeneous, i.e. that $\gamma_{ij}$ is just a constant tensor, we can easily derive the corresponding modification to the dispersion relations of magnons. Once again, the case of ferro- and ferri-magnets need to be treated separately from the case of antiferromagnets, for which $c_2 = 0$. The final outcome is that the magnon dispersion relations retain the same qualitative form, but the parameters $\Delta, m$ and $v_\chi^2$ get modified as follows:

$$\Delta \to \Delta' = \Delta \left[ 1 + \gamma \left( 1 - \frac{c_{10}}{c_6} \right) \right], \tag{58a}$$

$$m \to m' = m \left[ 1 + \gamma \left( 1 - \frac{c_8}{c_7} \right) - 2 \frac{c_9}{c_7} \gamma^{ij} \hat{k}_i \hat{k}_j \right], \tag{58b}$$

$$v_\chi^2 \to v_\chi^{2\prime} = v_\chi^2 \left[ 1 + \gamma \left( \frac{c_8}{c_7} - \frac{c_{10}}{c_6} \right) + 2 \frac{c_9}{c_7} \gamma^{ij} \hat{k}_i \hat{k}_j \right]. \tag{58c}$$

Interestingly, it remains true that $\Delta' = 2m' v_\chi^{2\prime}$. We should also emphasize that the full action (49) can also be used to calculate the magnon dispersion relations in regimes where $\gamma_{ij} \sim \mathcal{O}(1)$. In that case, however, one needs to take into account the full non-linear structure of the functions $F_i(B)$. The advantage of focusing on small strains is that the coefficients appearing in (57) will also control other phenomena, such as the magnetic damping we are about to discuss. The effect of straining the lattice on anti-ferromagnetic magnons has also been studied in [59].

## D. Magnetic Damping

As previously mentioned our analysis has not included the Gilbert damping, which is typically added as a phenomenological term, but for magnetic insulators the damping arises due to magnon decay mediated by the interaction Lagrangian in Eq. (55). The decay width can be calculated from the cut diagram, which is the square of the amplitude shown in Fig. 2. This process induces a torque on the lattice that contributes to the Einstein-de Haas effect [60]. The converse process, where a phonon emits a magnon, is not allowed unless some of the symmetries are explicitly broken, as will be discussed in the next section. For simplicity, in what follows we are going to focus on (anti-)ferromagnets. Our analysis can be easily extended to the case of ferrimagnets.

*Ferromagnets.* On general grounds, we would expect interactions with the lowest number of derivatives to give the dominant low-energy contribution to the process shown in Fig. (2). In ferromagnets, where $c_2 \neq 0$, this suggests that we focus on the term in the first line of Eq. (55). In fact, when the derivatives are estimated on-shell using the dispersion relation appropriate for ferromagnets, we find that

$$\frac{\frac{c_2}{2} \partial_i \pi^i \epsilon_{ab} \chi^a \partial_t \chi^b}{\frac{c_2}{2} \epsilon_{ab} \chi^a \partial_t \pi^i \partial_i \chi^b} \sim \frac{k^3/m}{v_\pi k^2} = \frac{k}{m v_\pi}. \tag{59}$$

This means that the second interaction in (55) is actually the leading one, i.e.

$$\mathcal{L}_{\text{int}} \to -\frac{c_2}{2} \epsilon_{ab} \chi^a \partial_t \pi^i \partial_i \chi^b. \tag{60}$$

The corresponding amplitude is given by

$$i\mathcal{M} = -\frac{i}{2\sqrt{c_1}} \omega_\lambda(k) \hat{\epsilon}_\lambda^\star(k) \cdot (\vec{p} + \vec{p}') \tag{61}$$

where $\omega_\lambda(k)$ and $\hat{\epsilon}_\lambda(k)$ are respectively the dispersion relation and the polarization vector associated with a phonon of polarization $\lambda$. Notice also that the amplitude associated with the interaction (60) includes a factor of $(1/\sqrt{c_1})(1/\sqrt{c_2})^2$ that accounts for the non-canonical normalization of the phonon and magnon fields.

The total decay rate can be obtained as usual by integrating the amplitude squared over all possible final states that conserve momentum, with a relativistic (nonrelativistic) normalization for the phonon (magnon) states. The explicit results for longitudinal and transverse phonons are:

$$\Gamma_L = \frac{1}{4c_1} \int \frac{d^3 p'}{(2\pi)^3} \frac{d^3 k}{2\omega_L(k)(2\pi)^3} \omega_L(k)^2 \frac{[(\vec{p}+\vec{p}\,') \cdot \vec{k}]^2}{\vec{k}^2} (2\pi)^4 \delta^3(\vec{p}-\vec{p}\,'-\vec{k})\delta(\omega(p)-\omega(p')-\omega_L(k))$$
$$= \frac{2m^3 v_L^3}{3\pi\bar{\rho}p}(p-mv_L)^3 \theta(p-mv_L), \tag{62}$$

and

$$\Gamma_T = \frac{1}{4c_1} \int \frac{d^3 p'}{(2\pi)^3} \frac{d^3 k}{2\omega_T(k)(2\pi)^3} \omega_T(k)^2 \left\{ (\vec{p}+\vec{p}\,')^2 - \frac{[(\vec{p}+\vec{p}\,') \cdot \vec{k}]^2}{\vec{k}^2} \right\} (2\pi)^4 \delta^3(\vec{p}-\vec{p}\,'-\vec{k})\delta(\omega(p)-\omega(p')-\omega_T(k))$$
$$= \frac{mv_T}{15\pi\bar{\rho}p}(p-mv_T)^4 \left(4p+mv_T\right) \theta(p-mv_T), \tag{63}$$

where in final results we have used the fact that $c_1$ is equal to the background density $\bar{\rho}$.

*Anti-Ferromagnets.* In the antiferromagnetic case, $c_2 = 0$ and the power counting is such that the momentum and energy scale in the same way. This is because both phonons and magnons now have linear dispersion relations: $\omega_{L,T}^2 = v_{L,T}^2 k^2$ and $\omega^2 = v_\chi^2 p^2$, respectively. Thus all the terms in (55) contribute at the same order, and the expressions for the decay rates become more complicated:

$$\Gamma_T = \frac{2p^5(1-\hat{v}_T)\hat{v}_T \left(\hat{v}_T^3 + 6\hat{v}_T^2 + 14\hat{v}_T + 14\right)(c_6+c_9)^2}{105\pi c_1 c_6^2(\hat{v}_T+1)^5} \Theta(1-\hat{v}_T), \tag{64}$$

$$\Gamma_L = \frac{p^5}{210\pi c_1 c_6^2 \hat{v}_L (\hat{v}_L+1)^5} (4c_6^2\hat{v}_L^6 + 20c_6^2\hat{v}_L^5 + 32c_6^2\hat{v}_L^4 - 56c_6 c_{10}\hat{v}_L^2 + 8c_6 c_8\hat{v}_L^4 + 40c_6 c_8\hat{v}_L^3 + 8c_6 c_8\hat{v}_L^2 + 8c_6 c_9\hat{v}_L^6$$
$$+40c_6 c_9\hat{v}_L^5 + 72c_6 c_9\hat{v}_L^4 + 40c_6 c_9\hat{v}_L^3 - 48c_6 c_9\hat{v}_L^2 + 14c_{10}^2\hat{v}_L^2 - 35c_{10}^2\hat{v}_L + 35c_{10}^2 + 28c_{10}c_8\hat{v}_L^2 - 70c_{10}c_8\hat{v}_L + 14c_{10}c_8$$
$$-140c_{10}c_9\hat{v}_L + 84c_{10}c_9 + 18c_8^2\hat{v}_L^2 - 15c_8^2\hat{v}_L + 11c_8^2 + 8c_8 c_9\hat{v}_L^4 + 40c_8 c_9\hat{v}_L^3 + 72c_8 c_9\hat{v}_L^2 - 100c_8 c_9\hat{v}_L + 36c_8 c_9 + 4c_9^2\hat{v}_L^6$$
$$+20c_9^2\hat{v}_L^5 + 40c_9^2\hat{v}_L^4 + 40c_9^2\hat{v}_L^3 + 12c_9^2\hat{v}_L^2 - 120c_9^2\hat{v}_L + 60c_9^2)\Theta(1-\hat{v}_L), \tag{65}$$

where $\hat{v}_{L,T} \equiv v_{L,T}/v_\chi$ .

The purpose of this calculation is only illustrative. For one thing the result is a function of the unknown quantities $(v_{L,T}, v_\chi, c_6, c_8, c_9, c_{10})$, all of which would have to be fit from data. Furthermore, phenomenologically, one would typically be more interested in the finite temperature decay rate as as well as the transport lifetime. This analysis was performed for the special case of Yttrium Iron Garnet in [46]. It is straightforward exercise to calculate these quantities in the effective field theory.

## VII. EXPLICIT SYMMETRY BREAKING

Explicitly breaking internal spin rotations leads to a broad range of interesting phenomena. To gain some physical intuition for how explicit symmetry breaking can arise, we shall begin by recalling the microscopic origin of the symmetric Lagrangian in the incompressible limit, Eq. (37).

### A. Continuum limit of the Heisenberg model

The strong coupling expansion of the half filled Hubbard model reduces to the Heisenberg model,

$$H = -J \sum_{\langle ij \rangle} \vec{\mathcal{S}}_i \cdot \vec{\mathcal{S}}_j. \tag{66}$$

Since the Hubbard model only involves spin independent nearest neighbor interactions, this Hamiltonian is independent of the magnetic moment. i.e. $J$ only depends upon the matrix element of the Coulomb interaction between electrons centered on neighboring atoms. In this way we can think of the Heisenberg model as an effective theory of the Hubbard model where we have integrated out the atomic orbits. At higher orders in the strong coupling expansion, the Hamiltonian (66) gets corrected by the so-called "bi-quadratic" terms of the form

$$\Delta H = -\tilde{J} \sum_{\langle ij \rangle} (\vec{\mathcal{S}}_i \cdot \vec{\mathcal{S}}_j)^2. \tag{67}$$

While such terms, if numerically significant, can have considerable effects on the phase transition [61] the low energy theory of Goldstones below the critical point is unchanged by their presence.

Starting from the Heisenberg Hamiltonian (66), we can obtain (minus) the static limit of the Lagrangian density (40) by taking to the continuum limit. This is accomplished by parameterizing the spins as $\vec{\mathcal{S}}_i \equiv \mathcal{S}\hat{n}_i$, where the magnitude $\mathcal{S}$ is constant and replacing $i \to \vec{r}, j \to \vec{r} + \vec{\delta}$, where $\vec{r}$ is the position of the $i$th spin with some choice of origin. The sum over nearest neighbors becomes an integral over $\vec{r}$. We then coarse grain by averaging over the $\vec{\delta}$'s,[12] and take the limit $\vec{\delta} \to 0, \mathcal{S} \to \infty$ with $\delta^2\mathcal{S}^2$ fixed.

The final result is

$$-\mathcal{L}_{\text{static}} = \frac{c_7}{2}(\partial_i\hat{n})^2, \qquad (68)$$

and $c_7 \sim J\delta^2\mathcal{S}^2$.

## B. Explicit symmetry breaking and spurions

To properly capture the long distance physics of explicit symmetry breaking we utilize a spurion analysis (see e.g. [62]). We will assume that the associated length and time scales are much longer than those at which spontaneous symmetry breaking occurs, so that explicit breaking can be treated perturbatively using spurion fields. The symmetry breaking parameter (in cut-off units) is treated as an additional expansion parameter, whose relative size compared to other corrections will depend upon the energy/length scale of interest.

### 1. Zeeman Interactions

Arguably the simplest source of explicit symmetry breaking is the Zeeman coupling between spins and a constant external magnetic field. At the microscopic level, this is described by supplementing the microscopic Hamiltonian with a term

$$\Delta H = -\mu \sum_i \vec{\mathcal{B}} \cdot \vec{\mathcal{S}}_i. \qquad (69)$$

This interaction explicitly breaks the spin $SO(3)$ down to the $SO(2)$ subgroup that leaves $\vec{\mathcal{B}}$ invariant

The spurion technique amounts to treating the explicit symmetry breaking as if it were a *spontaneous* breaking due to an operator $\vec{\Psi}$—the spurion field—that develops a small expectation value $\langle\vec{\Psi}\rangle = \mu\vec{\mathcal{B}}$. The advantage of this approach is that the spurion can be treated like any other matter field and coupled to the Goldstone modes following the standard rules of the coset construction [16, 63]. The spurion transforms in a linear representation of the full symmetry group $(G)$, $\vec{\Psi} \to g\vec{\Psi}$. However, to form invariant using the coset construction we are interested in objects which transform under the unbroken subgroup $H$. The field $\vec{\Psi}' = \Omega^{-1}\vec{\Psi}$ is such an object as it transforms as $\vec{\Psi}' \to h(\Phi, g)\vec{\Psi}'$, where $\Phi$ stands for all the Goldstone fields. However, $\vec{\Psi}'$ transforms reducibly under $H$ so we decompose $\vec{\Psi}'$ into irreducible representations of the unbroken group, i.e. $\Psi'_a$ and $\Psi'_3$. Finally we add to the effective action terms that depend on these irreps and are manifestly invariant under the unbroken group. To this end it is helpful to notice that the microscopic interaction preserves time reversal if the spurion is assumed to be odd, i.e. to transform as $\vec{\Psi} \to -\vec{\Psi}$.

In a ferromagnet, where time reversal is spontaneously broken, we are allowed to write terms involving the spurion that are not invariant under time reversal. Consequently, at leading order in $\mu\mathcal{B}L_\chi/v_\chi$ we have

$$\mathcal{L}_{\text{spurion}} = F(B)\,\Psi'_3 = F(B)\,O_{3A}^{-1}(\chi)\Psi^A = F(B)\hat{n}\cdot\vec{\Psi} \to F(B)\mu\hat{n}\cdot\vec{\mathcal{B}}, \qquad (70)$$

where in the last step we have replaced the spurion with its expectation value. Since, in the continuum limit, an external magnetic field couples to the Noether density of spin [22], the function $F(B)$ is constrained [13]. More precisely, since the Ferromagnetic spin density is given by $\vec{s} = c_2\det(D)\hat{n}$ for a ferromagnet, this fixes $F(B) = c_2\det(D)$. The operator in (70) introduces mixing between magnons and longitudinal phonons when $\vec{\mathcal{B}}$ is not aligned with the unbroken spin direction (the 3 direction, in our notation)[14] . Of course, the incompressible limit ($F(B) = \text{constant}$) of this result could have also been obtained more easily by taking the continuum limit of the microscopic interaction (69).

---

[12] By isotropy, we must have $\langle\delta_i\delta_j\rangle \sim \delta^2\delta_{ij}$.

[13] We thank Tomas Brauner for pointing this out to us.

---

[14] When $\vec{\mathcal{B}}$ is not aligned with the magnetization, the system will precess around the field. Damping will eventually lead to alignment on longer time scales.

In the case of an antiferromagnet, the leading interaction with the spurion must be invariant under time reversal, and therefore we have

$$\mathcal{L}_{\text{spurion}} = F(B)\,\Psi'_a\nabla_t\chi^a \to F(B)\,O^{-1}_{aA}(\chi)\,\mu\mathcal{B}^A\nabla_t\chi^a. \tag{71}$$

Of course, the interaction (71) is also allowed for ferromagnets. But in the incompressible limit, this is not the leading correction to the effective action for ferromagnons. The functional form of $F(B)$ is also constrained in this case from the anti-ferromagnetic spin density to be $\rho\tilde{F}_3(B)$. As in (70), this also results in phonon-magnon mixing when $\vec{\mathcal{B}}$ is not aligned with the unbroken spin direction. Interestingly, Zeeman interactions cannot introduce mixing between magnons and transverse phonons—a result that follows straightforwardly from our spurion analysis.

### 2. The Dzyaloshinsky-Moriya (DM) interactions

At the microscopic level the Dzyaloshinsky-Moriya (DM) interaction [64, 65] takes the form:

$$H = \sum_{\langle ij\rangle}(\vec{\mathcal{S}}_i\times\vec{\mathcal{S}}_j)\cdot\vec{D}_{ij}, \tag{72}$$

where the vector $\vec{D}_{ij}$ depends on two neighboring lattice points, and in perturbation theory can be expressed as a linear combination of matrix elements of the orbital angular momentum operator [66]. This interaction occurs when the inversion symmetry is broken in a material, and leads to the canting of the spins in the ground state. It explicitly breaks spin and spatial rotations down to the diagonal subgroup, generated by $\vec{J}\equiv\vec{S}+\vec{L}$.

At the microscopic level, one can distinguish between two types of DM interactions depending on whether $\vec{D}_{ij}$ is parallel or perpendicular to the lattice vector $\vec{r}_{ij}$ connecting the sites $i$ and $j$. In the continuum limit, the first case yields the so-called *Bloch-type* DM interactions, which arise for instance in non-centrosymmetric bulk materials [67]. In the second case, the resulting DM interaction is dubbed *Néel-type*. This interaction is anisotropic, and it occurs for example when a thin film ferromagnet is placed on top of a non-magnetic material with a large spin-orbit interaction (interfacial DM interaction) [68]. Significant theoretical and experimental attention has been recently devoted to DM interactions, as they provide a mechanism to stabilize magnetic Skyrmions [69–82].

Instead of taking the continuum limit of the microscopic interactions (72), we are going to use the spurion technique to infer the corresponding terms in the effective action for magnons and phonons. In order to break spatial and spin rotations down to the diagonal subgroup, we need a spurion field that transforms in a non-trivial representations of both symmetries, which we will take to be the fundamental representations for simplicity, i.e.

we will use a field $\Psi^A_i$. There are two distinct ways of implementing the desired explicit breaking by giving a vev to the spurion, and they correspond to the two types of DM interactions mentioned above:

$$\text{Bloch:}\quad \langle\Psi^A_i\rangle = \delta^A_i D_\parallel, \tag{73a}$$

$$\text{Néel:}\quad \langle\Psi^A_i\rangle = \epsilon^A{}_{ij}D^j_\perp. \tag{73b}$$

In order to couple the spurion to phonon and magnons, we will follow the blueprint outlined for the Zeeman interaction: we first introduce a new field $\Psi'\equiv\Omega^{-1}\Psi$, then break it up into its irreducible representations under the (spontaneously) unbroken group, $\Psi'^3_i$ and $\Psi'^a_i$. The leading symmetry breaking term in the effective Lagrangian is then

$$\mathcal{L}_{\text{spurion}} = F(B)\,\Psi'^a_i\nabla^i\chi_a \to F(B)\,O^{-1}(\chi)^a{}_A\langle\Psi^A_i\rangle\nabla^i\chi_a. \tag{74}$$

It is easy to show that, after replacing the spurion with the appropriate expectation values in (73) and taking the incompressible limit ($F(B) = \text{constant}$), this spurion action reproduces the familiar expressions for the Bloch and Néel DM interactions:

$$\text{Bloch:}\quad D_\parallel\epsilon_{ijk}\hat{n}^i\partial^j\hat{n}^k, \tag{75a}$$

$$\text{Néel:}\quad D^j_\perp(\hat{n}_j\partial_i\hat{n}^i - \hat{n}^i\partial_i\hat{n}^j). \tag{75b}$$

Away from the incompressible limit, the coupling (74) gives rise to a kinetic mixing between the longitudinal phonon and either $\partial_a\chi^a$ (Bloch) or $\epsilon^{ab}\partial_a\chi_b$ (Néel). This however is not the only source of kinetic mixing, since one should also consider the operator

$$\mathcal{L}'_{\text{spurion}} = F'(B)\,\Psi'^a_i\nabla^{(i}\pi^{j)}\nabla_j\chi_a. \tag{76}$$

which additionally generates a kinetic mixing between magnons and the transverse phonons. See e.g. [83–86] for recent work on phonon-magnon mixing.

## VIII. CONCLUSIONS

We have demonstrated how to build an effective field theory for magneto-elastic interactions using the space-time coset construction. The action non-linearly realizes all of the broken symmetries in a long wavelength approximation. The action includes all orders in the fields with a fixed number of derivatives, which makes the theory valid for any background where $\partial^2\chi/\Lambda^2_\chi \ll 1$. We have also shown how to systematically include the effects of explicit symmetry breaking due to Zeeman and DM interactions. Other symmetry breaking terms can be included using the same line of reasoning as presented in the last section. We have presented several new results most important of which are eqs. (50) and (51) that generalized the Landau-Lifshitz equations to allow for incompressibility. Applications of our formalism to

Skyrmionic physics will follow in a subsequent publication.

**Acknowledgments:** We would like to thank Amit Acharya, Tomas Brauner, Angelo Esposito, Garrett Goon and Di Xiao for helpful discussions. R.P. acknowledges the hospitality of the Sitka Sound Science Center and the Abdus Salam International Center for Theoretical Physics, where part of this work was carried out. This work was partially supported by the US Department of Energy under grants DE- FG02-04ER41338 and FG02-06ER41449, and by the National Science Foundation under Grant No. PHY-1915611.

## Appendix A: WZW term for magnons

In this short appendix, we provide a few more details about the derivation of the RHS of Eq. (10b). To this end, we'll focus our attention on the 2-form $\omega_2 \equiv \epsilon_{ab}\, \omega_{S_a} \wedge \omega_{S_b}$, which can be written more explicitly as

$$\omega_2 = \tfrac{1}{2}\epsilon^{aBC}[O^{-1}dO]_{a3} \wedge [O^{-1}dO]_{BC}. \tag{A1}$$

Using the fact that $O^{-1} = O^T$, and writing explicitly the sums over the indices $B = b, 3$ and $C = c, 3$, we find

$$\omega_2 = \epsilon^{ab} O_{Aa} O_{Bb}\, dO_{A3} \wedge dO_{B3}. \tag{A2}$$

At this point, it is convenient to think of the matrix elements $O_{AB}$ as a triplet of mutually orthogonal unit vectors defined by

$$\hat{m}_A^{(a)} \equiv O_A{}^a, \qquad \hat{n}_A \equiv O_{A3} \tag{A3}$$

Then,

$$\epsilon^{ab} O_{Aa} O_{Bb} = (\hat{m}_A^{(1)}\hat{m}_B^{(2)} - \hat{m}_A^{(2)}\hat{m}_B^{(1)}) = \epsilon_{ABC}\hat{n}^C. \tag{A4}$$

The result on the RHS follows from the fact that the expression in the intermediate step must be antisymmetric, orthogonal to $\hat{n}^A$ and $\hat{n}^B$, and its contraction with $\epsilon^{ABC}\hat{n}_C$ must be equal to 2. Thus, the 2-form in Eq. (A2) can be written as

$$\omega_2 = \epsilon_{ABC}\, \hat{n}^A\, d\hat{n}^B \wedge d\hat{n}^C. \tag{A5}$$

Now, if we parametrize the unit vector as in Eq. (39), we can calculate $\omega_2$ explicitly to obtain

$$\omega_2 = 2\sin\theta\, d\theta \wedge d\phi = d[-2\cos\theta\, d\phi]. \tag{A6}$$

However, the discussion in Sec. V A shows that this is also equivalent to

$$\omega_2 = d[\epsilon^{ab}(O^{-1}dO)_{ab}]. \tag{A7}$$

## Appendix B: Magnons in ferromagnets

In the ferromagnetic case, the term in the action with one time derivative is the leading order kinetic term. Therefore, we may eliminate the term with two time derivatives via a field redefinition such that

$$\partial_t\chi_a \partial_t \chi^a \to (\partial^2\chi_a)^2 + \dots. \tag{B1}$$

where the remaining terms involves sub-leading operators (see e.g. [41]). Recall that our power counting for the FM case dictates that time derivatives scale like two spatial derivatives, based on the dispersion relation $\omega = k^2/2m$. Then, the effective action for a ferromagnet describes a single propagating degree of freedom. This can be traced back to the existence of a primary (second class) constraint

$$p_\chi^a - \frac{1}{2}\epsilon^{ab}\chi^b = 0, \tag{B2}$$

where the $p_\chi^a$'s are the momenta conjugate to the $\chi_a$'s. The canonical quantization of this constrained theory has been discussed in detail in [25, 87]. One must use care in defining the external states, by proceeding through the Dirac procedure for constrained systems. The Dirac bracket algebra will be satisfied via the field expansions for $\chi$ and its conjugate momentum $p_\chi$,

$$\chi^a = \int \frac{d^3k}{(2\pi^3)}(a_k\epsilon^a e^{-ik\cdot x} + a_k^\dagger \epsilon^{a\star}e^{ik\cdot x})$$

$$p_\chi^a = -\frac{1}{2}\int \frac{d^3k}{(2\pi^3)}(a_k\epsilon^a e^{-ik\cdot x} - a_k^\dagger \epsilon^{a\star}e^{ik\cdot x})$$

$$\tag{B3}$$

where $k\cdot x = -\omega_k t + \vec{k}\cdot\vec{x}$ and $[a_k, a_k^\dagger] = (2\pi)^3\delta^3(\vec{k} - \vec{k}')$, and

$$\epsilon^a = (1, -i)/\sqrt{2}. \tag{B4}$$

This is equivalent to the statement that the complex field $\Psi = \frac{1}{\sqrt{2}}(\chi_1 + i\chi_2)$ only contains annihilation operators, as is the case for an ordinary non-relativistic field:

$$\Psi = \int \frac{d^3k}{(2\pi^3)}\, a_k e^{-ik\cdot x} \tag{B5}$$

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
