# Peer review of "An Effective Field Theory of Magneto-Elasticity"

_SciPost Physics_

## Round 2 · Referee Report · Anonymous · 2022-2-12

Strengths
1. The paper was well-written and easy to follow.
2. The purpose of the paper was clear from the beginning and all calculations were carried out methodically to support the initial purpose.
3. The topic of the paper was interesting and has clearly opened avenues of future research.
4. Not only did the authors treat the subject of magnons in solids from a novel perspective, they also derived new, more general equations governing their behavior.
Weaknesses
1. None
Report
This journal's acceptance criteria are met and I recommend this paper for publication.
Requested changes
1. Before submitting the final draft, check for typos and misspellings. I found a few along the way.

---

## Round 2 · Referee Report · Anonymous · 2022-2-19

Report
This manuscript provides a systematic construction of zero-temperature effective actions for the Goldstone modes present in solid ferromagnets, anti-ferromagnets, and ferrimagnets. This is achieved in the framework of the coset construction: all of these systems are described by the same symmetry breaking pattern, varying only with respect to their time-reversal properties. The main result of the manuscript is the explicit presentation of these effective actions, to fixed order in the derivative expansion but to all orders in the fields. The authors then proceed to various checks and applications: they show to specialize to the separate cases of phonon and magnons Lagrangians (easily reproducing known results), they discuss aspects of magnon-phonon interactions including magnetic damping via phonon channels (which were previously only obtained by more phenomenological approaches), and show how one can introduce perturbatively small explicit symmetry breaking terms, leading to physically relevant actions for pseudo-Goldstone.
The paper is very well-written and contains non-trivial results. This systematic construction of magneto-elasticity can find applications to condensed matter systems, such as Skyrmions which the authors plan to study next. It would also be interesting to perform further loop computations which might contribute to experimentally relevant physical phenomena. And of course, this is the first step in the direction towards formulating a hydrodynamic theory of magneto-elasticity at finite temperature. So, there is a clear potential for further work based on the results presented here.
The logical steps, as well as the calculations, are clear and easy to follow, and I did not spot a single typo. The abstract, introduction and conclusion present a succinct overview of the achievements of the paper, and the bibliography contains the most relevant previous work in this wide research topic.
The only minor improvement that could be made is the addition of further comments on the physical implications of the results, as well as the future directions. The expert in the field can understand their significance, but it would be nice to mention some experimental setups in which the theory can be potentially applied, name other physical systems which might be relevant, as well as make informed guesses on novel phenomena in extending these results to the case of finite temperature.
All in all, the paper is timely, clearly written, and contains interesting results; so I would be very happy to suggest it for publication in SciPost regardless of whether further comments are added.

---

## Editorial Decision

unknown